# Targeted Treatment against Cancer Stem Cells in Colorectal Cancer

**DOI:** 10.3390/ijms25116220

**Published:** 2024-06-05

**Authors:** Julia Martínez-Pérez, Carlos Torrado, María A. Domínguez-Cejudo, Manuel Valladares-Ayerbes

**Affiliations:** 1Medical Oncology Department, Hospital Universitario Virgen del Rocio (HUVR), Avenida de Manuel Siurot s/n, 41013 Seville, Spain; julia0802@hotmail.com; 2Institute of Biomedicine of Seville (IBiS), Hospital Universitario Virgen del Rocio (HUVR), Consejo Superior de Investigaciones Científicas, Universidad de Sevilla, Avenida de Manuel Siurot s/n, 41013 Seville, Spain; mdcejudo-ibis@us.es; 3Department of Investigational Cancer Therapeutics, The University of Texas MD Anderson Cancer Center, Houston, TX 77030, USA; ctorrado@mdanderson.org

**Keywords:** cancer stem cell, colorectal cancer, molecular targeted therapies, clinical trial

## Abstract

The cancer stem cell (SC) theory proposes that a population of SCs serves as the driving force behind fundamental tumor processes, including metastasis, recurrence, and resistance to therapy. The standard of care for patients with stage III and high-risk stage II colorectal cancer (CRC) includes surgery and adjuvant chemotherapy. Fluoropyrimidines and their combination with oxaliplatin increased the cure rates, being able to eradicate the occult metastatic SC in a fraction of patients. The treatment for unresectable metastatic CRC is based on chemotherapy, antibodies to VEGF and EGFR, and tyrosine-kinase inhibitors. Immunotherapy is used in MSI-H tumors. Currently used drugs target dividing cells and, while often effective at debulking tumor mass, these agents have largely failed to cure metastatic disease. SCs are generated either due to genetic and epigenetic alterations in stem/progenitor cells or to the dedifferentiation of somatic cells where diverse signaling pathways such as Wnt/β-catenin, Hedgehog, Notch, TGF-β/SMAD, PI3K/Akt/mTOR, NF-κB, JAK/STAT, DNA damage response, and Hippo-YAP play a key role. Anti-neoplastic treatments could be improved by elimination of SCs, becoming an attractive target for the design of novel agents. Here, we present a review of clinical trials assessing the efficacy of targeted treatment focusing on these pathways in CRC.

## 1. Introduction

Colorectal cancer (CRC) is a significant global health concern, ranking as the third most common cancer worldwide. Despite progress in screening techniques and therapeutic strategies, CRC continues to be a leading cause of cancer-related deaths globally [1]. The heterogeneity of CRC presents a challenge for treatment and contributes to its variable prognosis. This complexity is attributed to diverse genetic, epigenetic, and environmental influences. Colorectal cancer stem cells (CRC-SCs) have been identified as a critical factor in the initiation, progression, metastasis, and recurrence of the disease. The theory of cancer stem cells (CSCs) proposes that a small population of cells, the CSCs, may be responsible for the fundamental processes of cancer. This model suggests that there is a hierarchical organization within tumors, with CSCs at the top due to their unique self-renewal and multipotent properties, which can give rise to a heterogeneous population through asymmetric division. CSCs were first described in acute myeloid leukemia by Dick and colleagues [2]. They made a significant discovery that not all leukemic cells propagate the disease when transplanted into immunodeficient mice [2,3]. Later, using similar experimental approaches, the presence of CSCs was demonstrated in solid cancers such as CRC [4,5]. Human CRC-SCs were first identified based on cell surface expression of CD133. CD133+ cells were consistently able to reproduce the original tumor in immunodeficient mice, while their CD133-counterparts were unable to give rise to xenografts [4,5]. CD133+ cells were largely resistant to oxaliplatin and/or 5-fluorouracil (5-FU) in vitro and in vivo, whereas CD133− cells showed a high sensitivity to the treatment [6]. Today, it is well accepted that CRC-SCs represent a phenotypically and functionally heterogeneous population. Accordingly, several SCs markers have been reported including CD44, EpCAM, CD166, leucine-rich repeat-containing G-protein coupled receptor 5 (LGR5), aldehyde dehydrogenase 1 (ALDH1), EphrinB receptors, doublecortin-like kinase 1 (DCLK1), and others revised in [7].

CRC-SCs have two main sources: SCs that have undergone malignant transformation through genetic or epigenetic alterations and differentiated epithelial non-SCs that acquire stemness through dedifferentiation [6,8,9,10,11]. The dual origin of SCs (Figure 1) underscores their complexity and plasticity, as they possess self-renewal capacity and potential to drive tumor growth, metastasis, and resistance to conventional therapies [9]. The transformation of normal colonic SCs into CSCs is believed to be influenced by deregulation in several key signaling pathways, including Wingless integrated-Beta-catenin (Wnt/β-catenin), Hedgehog, Notch, and TGF-β/SMAD (Figure 1). The dedifferentiation of non-stem epithelial cells into tumor-initiating cells depends mainly on the level of Wnt signaling and is also influenced by the tumor microenvironment (TME) through different signaling pathways, such as PI3K/Akt/mTOR, JAK/STAT, and NF-κB, which link inflammatory and immune signaling, angiogenesis, and tumor-initiating cells [11]. Recent advances have revealed a distinct SC type within CRC dynamics, termed revival colon SCs (revCSCs). In contrast to traditional proliferative CSCs, which prioritize proliferation, revCSCs emphasize survival over proliferation [12]. These SCs share transcriptional similarities with fetal intestinal SCs, suggesting the activation of early developmental programs in response to epithelial stress. Interestingly, revCSCs, which are slow-cycling cells, show a marked upregulation of Yes-associated protein (YAP) signaling. This pattern is complemented by increased signaling in pathways such as focal adhesion kinase (FAK), TNF-α, TGF-β, INF-γ, and NF-κB, which play a fundamental role not only in the initial formation of the primary tumor but also in metastatic dissemination and the emergence of chemoresistance [13].

## 2. Targeted Drug Modulation of Stemness Pathways

Several signaling pathways are involved in both the development and maintenance of CRC-SCs. This review aims to focus on clinical trials investigating the potential of targeted drug modulation of stemness pathways, including Wnt/β-catenin, Hedgehog, Notch, TGF-β/SMAD, PI3K/Akt/mTOR, NF-κB, Janus kinase/signal transducer and activator of transcription (JAK/STAT), DNA damage response, and Hippo-YAP, in the treatment of CRC (Figure 1). The evaluation of immunotherapy strategies against CSCs is outside the scope of this review.

### 2.1. Wingless Integrated/Beta-catenin (Wnt/β-catenin) Pathway

The Wnt/β-catenin pathway is a crucial molecular pathway involved in cell development, proliferation, and differentiation and plays a key role in embryonic development and adult tissue homeostasis. Wnt/β-catenin signaling plays a key role in influencing cancer stemness [14,15]. Dysregulation of this pathway is common in cancers and more than 80% of CRC cases are associated with driver mutations in this pathway [16,17]. Sequence variations in the Wnt pathway components, such as loss of function mutations in adenomatous polyposis coli (APC) and activating mutations in β-catenin, significantly influence the development and progression of CRC [17]. 

Therapeutic strategies that aim to disrupt the Wnt signaling pathway can reduce the population of CSCs and inhibit tumor growth. Currently, there are drugs approved by the Food and Drug Administration (FDA) for other medical conditions that target the Wnt/β-catenin pathway, including niclosamide (a Frizzled receptor inhibitor), pimozide (a β-catenin inhibitor), and ethacrynic acid (a β-catenin-LEF1 inhibitor). Additionally, preclinical data support the use of these agents in various types of cancer, including CRC [17]. Niclosamide is currently undergoing evaluation in an ongoing phase 2 trial for metastatic CRC (clinicaltrials.gov identifier NCT02519582, accessed on 26 April 2024). 

Regorafenib is a multitargeted kinase inhibitor approved for the treatment of patients with metastatic CRC (mCRC) who are refractory to standard chemotherapy [18]. Regorafenib treatment was shown to reduce the formation of tumor spheres and the side population in 5-FU-resistant CRC cell lines both in vitro and in immunocompromised mice. The stemness genes including Notch1, WNT1, CD44, and c-Myc were downregulated by regorafenib exposure via miR-34a induction [19]. A preclinical study has suggested that regorafenib may be useful in the adjuvant treatment of micrometastases in resected CRC [20]. However, a phase 3 trial that evaluated the efficacy and safety of regorafenib vs. placebo in patients with CRC following curative resection of liver metastases after completion of planned chemotherapy was prematurely terminated [21]. Only five patients were randomized. Serious adverse events (SAEs) were reported on 14.29% of the patients assigned to regorafenib [21]. A randomized phase 2 study is currently recruiting to evaluate the efficacy of durvalumab plus regorafenib compared to observation in patients with stage IV colon cancer who have achieved no evidence of disease following locoregional or systemic treatment. It is planned to enroll 182 mCRC patients. The primary endpoint is disease-free survival (NCT05382741).

Previous studies have shown that vitamin D can repress Wnt/β-catenin signaling in cancer cells [22]. In a phase 2 clinical trial involving 139 patients with mCRC, the addition of high-dose vitamin D3 to standard chemotherapy was compared to standard-dose vitamin D3. The study found a difference in median progression-free survival (PFS) between the two groups (13 vs. 11 months), although this difference was not statistically significant. No significant differences were found between high-dose and standard-dose vitamin D3 for objective response rate (ORR, 58% vs. 63%; *p* = 0.27) or overall survival (OS, median 24.3 months vs. 24.3 months, respectively; *p* = 0.43) [23].

Genistein is a glycogen synthase kinase-3 beta (GSK-3 β) inhibitor that blocks Wnt signaling by increasing the production of soluble Wnt inhibitory molecules. In a phase 1/2 pilot study, genistein was combined with FOLFOX or FOLFOX–bevacizumab for the first-line treatment of 13 mCRC patients. The study showed a favorable safety profile, with an ORR of 61.5% and a median PFS of 11.5 months (95% confidence interval, CI 4.9–21.7) [24]. Another selective GSK-3 β inhibitor, 9-ING-41, was studied as a single agent or with chemotherapy in a phase 1b/2 trial involving 63 patients with solid tumors. Only one of the thirteen patients with CRC who were treated achieved SD [25].

Various targeted therapies against porcupine, an enzyme that facilitates the acylation of Wnt proteins, have reached different phases of development in CRC. In a phase 1b trial, a small molecule inhibitor of porcupine O-acyltransferase (WNT974) did not show any clinical benefit when combined with encorafenib and cetuximab in BRAF V600E mutant patients. The trial observed dose-limiting toxicities (DLTs) in 4 out of 20 patients, and the overall response rate (ORR) was only 10%. As a result, a phase 2 trial was not started [26]. RXC004, another porcupine inhibitor, was evaluated in a clinical trial with 20 patients and was found to be safe at doses up to 2 mg. It is of note that RXC004 demonstrated disease stabilization in five patients with Wnt ligand-dependent tumors. It is currently being evaluated in a phase 2 trial for metastatic microsatellite stable CRC, either alone or in combination with nivolumab [27]. Tumors must have loss-of-function RNF43 mutation or R-spondin (RSPO) 2/3 fusion for enrollment.

Other target therapy approaches are under early stages of development for CRC treatment such us β-catenin inhibitors, RSPO antibodies, and later-generation β-catenin and frizzle receptor inhibitors [28].

### 2.2. Hedgehog Signaling Pathway

The Hedgehog signaling (Hh) pathway plays a crucial role in embryonic development and tissue regeneration. The Hh pathway has been linked to carcinogenesis in CRC and is also involved in maintaining stemness and drug resistance [29]. Smoothened (SMO) is a family of seven transmembrane proteins that are critical for Hh signal transduction on the cell membrane. Drugs targeting SMO have been shown to be the most effective way to modulate the activity of the Hh pathway.

Several SMO inhibitors (such as vismodegib or sonidegib) have been approved for treating basal cell carcinoma or hematological malignancies. Other novel SMO inhibitors are still in early development phases, revised in [30]. Hh pathway inhibitors have limited activity in unselected patients with solid tumors. 

In the context of CRC, certain compounds have demonstrated efficacy in preclinical models, but their translation to the clinic has not been successful. In a phase 2a multiple basket study, patients with advanced refractory solid tumors and molecular alteration in the Hh pathway were treated with vismodegib, although no patients with CRC were found [31]. In another phase 2 trial, patients with previously untreated mCRC were randomized to receive either vismodegib or a placebo in combination with a standard chemotherapy regimen. The results showed that the vismodegib arm had an equivalent ORR and similar median PFS compared to the placebo arm, indicating a lack of efficacy [32].

While the Hh pathway is a promising target for cancer treatment, a better understanding of the mechanisms that cause aberrant Hh activation and its consequences in different tumor types is necessary for the development of new therapies.

### 2.3. Notch Pathway

The Notch signaling pathway is essential for cell fate decisions and contributes to the transformation process by inhibiting the differentiation of SCs and promoting their proliferation, thereby maintaining the CSC phenotype. Alterations in this pathway have been associated with CRC [33]. High Notch1 copy number gains have been described in over 22% of CRCs [34] and other downstream molecular alterations have also been described in CRC patients [35].

Notch1 activation can be inhibited targeting gamma-secretase (γ-secretase), an enzyme responsible for cleaving Notch receptors. A phase 2 study evaluated the efficacy of RO4929097, a γ-secretase inhibitor, in 37 mCRC patients who had previously received at least two lines of chemotherapy. There were no radiologic responses. The median PFS was 1.8 months (95% CI, 1.8–1.86) and median OS was 6.0 months (95% CI, 3.9–9.1) [36]. Other γ-secretase inhibitors have been tested in CRC in phase 1, but with limited efficacy. Two out of six patients with CRC showed SD with the combination of RO4929097 with cediranib in a phase 1 trial evaluating a total of twenty patients with solid tumors [37]. In a phase 1 study evaluating LY9000009, another gamma-secretase inhibitor, two out of five patients with CRC showed SD [38].

Monoclonal antibodies (moAbs) targeting Notch are also under development. Demcizumab, an anti-delta ligand-4 (DLL4), had a favorable safety profile in a phase 1 study and showed disease stabilization in 40% of 55 evaluable patients, including 10 patients with CRC [39]. A phase 2 clinical trial is currently evaluating demcizumab in combination with FOLFIRI in first- and second-line mCRC, but results have not yet been reported (NCT01189942).

Other approaches, including ADAM inhibitors and Hes-1 inhibitors, are currently in preclinical or early phase 1 clinical trials [35].

### 2.4. TGF-β/SMAD Pathway

TGF-β is a cytokine with multifunctional properties that can act as a tumor promoter or tumor suppressor in a cell- and context-dependent manner. The TGF-β/SMAD pathway acts as a tumor suppressor in early stages and later facilitates transformation through mechanisms such as the epithelial–mesenchymal transition (EMT), which endows cells with stem-like features [40]. In the context of CRC, some studies with colonic organoids and xenograft models have demonstrated that the TGF-β signaling pathway is specifically activated in the CSCs in human CRC [41,42]. Moreover, increased levels in the primary tumor or high plasma TGF-β levels in patients are associated with poor prognosis [43]. 

Preclinical data suggest that TGF-β could be a promising therapeutic target in mCRC. Strategies to block TGF-β signaling have been proposed and several drugs have been evaluated in early phase clinical trials with limited success. In CRC, these drugs include receptor kinase inhibitor, moAbs, and ligand traps, either as monotherapy or in combination with other anti-cancer drugs [40,41,42].

Vactosertib is an oral inhibitor of TGF-β type I receptor kinase. A phase 1b/2a study was conducted to evaluate vactosertib in combination with pembrolizumab in 105 previously treated, immunotherapy-naïve patients with microsatellite stable (MSS) mCRC (NCT03724851). The ORR was 13.3% (95% CI, 7.5–21.4). Median PFS and OS were 1.3 months (95% CI, 1.2–1.4) and 15.8 months (95% CI, 7.9-NR), respectively. The phase 2 part of the study is still ongoing [44]. 

In a single-arm phase 2 trial (NCT02688712), galunisertib, another TGF-β kinase inhibitor, was tested in 38 previously untreated patients with locally advanced rectal cancer in combination with neoadjuvant chemoradiotherapy, showing a complete response rate of 32% [45].

NIS793, a fully human anti-TGF-β moAb, has shown promising results in a phase 1b study combined with spartalizumab in MSS-CRC (n = 40) [46]. NIS793 has also been evaluated in a phase 2 trial in combination with FOLFOX/FOLFIRI + bevacizumab + tislelizumab vs. chemotherapy for second-line treatment (NCT04952753) in mCRC. Although no results are available, Novartis will discontinue the development of the anti-TGF-β moAb NIS793 for the treatment of cancer. 

Also, the clinical results of ascrinvacumab (PF-03446962), a moAb against activin receptor-like kinase-1 (ALK1), a type I subclass of the TGF-β receptor with dose-dependent anti-angiogenic activity, have been disappointing in various tumor types [47,48]. In a study involving patients with chemorefractory mCRC (NCT02116894), the combination of PF-03446962 and regorafenib resulted in unacceptable toxicity without any evidence of anti-tumor activity, leading to the premature termination of the trial [49]. 

SAR439459, a next-generation anti-TGF-β antibody that inhibits all TGF-β isoforms, has been investigated against solid tumors. In 24 mCRC patients a disease control rate (DCR) of 25% and partial response (PR) of 4% was obtained. Nevertheless, the study was discontinued due to a lack of efficacy and a high risk of bleeding. No significant association between clinical response and plasma TGF-β level at baseline or modulation upon treatment was observed [50]. 

Bintrafusp alfa is a bifunctional fusion protein that targets PD-L1 and TGF-β. In CRC, bintrafusp alfa in monotherapy showed only modest anti-tumor activity. In an expansion cohort of a phase 1 trial (NCT02517398) evaluating bintrafusp alfa in monotherapy for heavily pretreated mCRC patients, the ORR was low (3.1%), as was the DCR (6.3%). Additionally, the median PFS was poor (1.3 months) and the median OS was 7.7 months [51]. Bintrafusp did not demonstrate significant anti-tumor activity in patients with high microsatellite instability (MSI-H) CRC who had previously received immunotherapy in a phase 2 trial with 15 patients. The median PFS was 1.8 months with no responses [52]. In a phase 2 trial, the combination with other immune-oncology agents (CV301—poxviral vaccine against CEA and MUC1, N-803—IL-15 superagonist, and M9241—tumor-targeted IL-12) showed preliminary clinical activity with a 12-month OS rate between 66.7% and 77.2% and a manageable safety profile in patients MSS CRC in the third or successive lines [53]. However, the phase 2 trial that evaluated the combination of bintrafusp with radiotherapy was terminated due to futility (NCT03436563).

SHR-1701 is a bifunctional anti-PD-L1/TGF-β agent. A phase 2 study combining SHR-1701 with standard therapy (XELOX and the anti-VEGF biosimilar BP102) in first-line treatment (NCT04856787) reported an ORR of 59.7% and a DCR of 83.9% in 62 mCRC patients. Median PFS was 10.3 months (95% CI, 8.3–13.7), with no mature OS data [54]. Grade ≥ 3 adverse events (AEs) were reported in 59.7% of patients with a 9.7% rate of discontinuation of SHR-1701, with anemia (8.1%) and decreased neutrophils (6.5%) being the most common toxicities.

Interference with EMT (directly or indirectly) and mesenchymal–epithelial transition (MET) has also been evaluated as a potential therapeutic strategy in preclinical models, with preliminary anti-tumor activity [55,56]. Nevertheless, further clinical research and additional studies are necessary before application in clinical practice.

### 2.5. PI3K/Akt/mTOR Pathway

The phosphoinositide 3-kinases (PI3Ks) are a family of activating enzymes that can phosphorylate the inositol ring in phosphatidylinositol membrane lipids. They are known to play a central role in cell growth, survival, metabolism, and cell mobility, which makes them a therapeutic target of great interest [57].

In models of colon cancer, a mutation in PIK3CA activates PI3K/Akt signaling and increases the expression of LGR5 and c-myc, promoting the survival and proliferation of stem cells. Tumor cells with a PIK3CA mutation and LGR5 expression exhibit poor sensitivity to first-line chemotherapy in vitro. In multivariate analysis, patients with a PIK3A mutation and LGR5 expression had poor outcomes for both DFS and OS [58]. PI3KCA mutations are present in 13% of mCRC cases and are associated with right-sided tumors, older patients, and poor prognosis [58,59,60].

Drugs that interfere with the PI3K pathway are currently undergoing clinical evaluation. These include PI3K inhibitors, dual inhibitors of PI3K and mTOR, AKT inhibitors, and mTOR inhibitors. PI3K inhibitors can be pan-inhibitors (such as buparlisib, pictilisib, copanlisib, pilaralisib, and sonolisib) or selective for the catalytic subunit p110 α (alpelisib, taselisib, inavolisib, and serabelisib). 

In a randomized phase 2 trial involving patients with mCRC, the combination of sonolisib and cetuximab did not demonstrate superiority in terms of PFS or OS when compared to cetuximab alone. The median PFS was 59 days for cetuximab and sonolisib and 104 days for cetuximab (*p* = 0.77). Similarly, the OS for arm combination and cetuximab alone were 266 and 333 days, respectively (*p* = 0.83). The treatment-related toxicity was higher in the combination arm, especially in terms of all-grade nausea, vomiting, diarrhea, and rash [61]. 

In a phase 1 study with 52 patients with advanced solid tumors, the combination of pilaralisib and paclitaxel–carboplatin was evaluated. The best overall response was confirmed PR in seven patients (13.5%). It included one CRC patient (no response). No associations between PI3K pathway alterations and responses were observed [62]. Buparlisib has been evaluated in a phase 1 trial in combination with mFOLFOX6 [63]. A total of 17 patients received treatment with buparlisib, 13 of which were evaluate for dose-limiting toxicities (DLTs). The most common tumor type was CRC. DLT included hyperglycemia, transaminase elevation, neutropenia, and thrombocytopenia. One patient demonstrated an unconfirmed PR and three patients had stable disease (SD).

In a phase 1b trial for patients with advanced CRC and KRAS wildtype, buparlisib was combined with panitumumab [64]. The toxicities observed included fatigue, rash, and mucositis. Out of the seventeen patients, one (5.9%) who was PTEN and PIK3CA negative by immunohistochemistry had a PR, seven had SD, and nine had disease progression. The authors conclude that in this biomarker-unselected cohort, there was limited evidence to suggest activity. 

The NCI-MATCH tumor-agnostic platform trial arm Z1F (NCT02465060) evaluated copanlisib in patients with PIK3CA mutations and KRAS wildtype. The study achieved its primary endpoint with an ORR of 16% (*p* = 0.0341). The most common toxicities observed were hyperglycemia, fatigue, diarrhea, hypertension, nausea, and rash. However, no clinical benefit (PR or SD) was observed in the four mCRC patients included [65]. 

Single-agent alpelisib has been evaluated in PIK3CA-altered advanced solid tumors (NCT01219699). Alpelisib in combination with fulvestrant is currently approved for the treatment of late-stage metastatic breast cancer in patients with a PIK3CA mutation. In the first dose-escalation and expansion study, 35 patients with mCRC (26.1%) were included. Two patients had PR. DCR and clinical benefit rate (CBR: complete response + PR + SD > 24 weeks) were 34.3% and 8.6%, respectively. Concomitant PIK3CA and KRAS mutations were detected in 13 of 17 (76.5%) colorectal tumors. In the complete cohort (n = 134) hyperglycemia, gastrointestinal (GI) toxicities, fatigue, and rash were the most frequent AEs [66].

In a randomized phase 2 study, the efficacy of adding alpelisib to encorafenib and cetuximab in previously treated BRAF-mutated mCRC patients was investigated [67]. The combination of cetuximab, encorafenib, and alpelisib resulted in a median PFS of 5.4 months and a confirmed ORR of 27% in 52 patients. The combination of encorafenib and cetuximab (n = 50) resulted in an mPFS of 4.2 months and an ORR of 22%. The triple combination showed favorable results for PFS, with an HR of 0.69 (0.43–1.11; *p* = 0.064).

There are other ongoing phase 1 trials in the treatment of CRC, evaluating alpelisib, capecitabine, and radiation in rectal cancer (NCT02550743) or alpelisib with capecitabine in mCRC with PIK3CA gene mutation (NCT04753203). 

AKT proteins are crucial effectors of PI3K and are directly activated in response to PI3K. One of the key downstream target genes of AKT is the mammalian target of rapamycin (mTOR) complex, which is a conserved serine/threonine kinase. Dual inhibitors of PI3K and mTOR are in early phases of drug development, such as dactolisib (BEZ235) [68].

In colon cancer SC models in vitro, Chen et al. reported that dactolisib reduced colon SCs proliferation, decreased expression of the stem cell markers CD133 and Lgr5, and induced apoptosis [69]. Dactolisib had been evaluated in a phase 1 study in 33 patients with advanced solid tumors. The AEs included mucositis, fatigue, hyperglycemia, dehydration, and thrombocytopenia. Fifteen patients experienced SD as their best response, including four colorectal patients who had disease control for ≥16 weeks [70]. In addition, a phase 1b dose-escalation study of everolimus plus dactolisib showed no responses in 19 patients with advanced tumors, including 3 CRC patients [71]. 

Another PI3K/mTOR inhibitor, gedatolisib, has also been evaluated in combination with irinotecan in a dose-escalation study [72]. The most common treatment-related AEs were nausea, diarrhea, vomiting, and mucosal inflammation/stomatitis. The ORR was 4.7% (95% CI, 0.6–15.8). Clinical benefit was seen in seven patients (16.3%; 95% CI, 6.8–30.7%). The median PFS was 2.8 (95% CI, 1.7–3.7) months. There was no correlation between clinical benefit and the presence of activating PI3K mutations [73]. 

In preclinical models, it has been observed that the overexpression of ABCB1 and ABCG2 in CRC cells, as well as aberrant regulation of WNT/β-catenin and active glycogen synthase kinase 3-beta (GSK3β) signaling, may lead to resistance to gedatolisib [28,74]. While the initial results are promising, there are still many issues associated with the use of PI3K pathway inhibitors in CRC that need to be resolved. These include selecting the appropriate biomarkers of pathway activation, determining the best timing for drug use, identifying resistance mechanisms, and investigating whether combination therapies can improve treatment outcomes.

### 2.6. NF-κB Pathway

NF-κB transcription factors play a crucial role in various physiological processes, including innate and adaptive immune responses, cell proliferation, cell death, and inflammation. These factors can contribute to the transformation of normal colonic SCs into CSCs [75]. Moreover, NF-κB signaling in the tumor microenvironment can induce dedifferentiation of cancer cells, enhancing their stemness and resistance to therapies. A constitutive activation of the NF-κB pathway has been described in 60 to 80% of CRCs [76]. 

The NF-κB pathway is activated through ubiquitin-mediated protein degradation via the proteasome. Although preclinical studies have shown promise for the development of proteasome inhibitors in CRC [77], a phase 2 study found that single-agent bortezomib, an FDA-approved proteasome inhibitor, was ineffective in treating mCRC [78]. None of the 19 patients with CRC who received bortezomib showed an objective response, and the median time to progression was 5.1 weeks [78].

Curcumin is a natural compound that inhibits the NF-κB pathway as well as other mechanisms, such as the WNT/β-catenin pathway [79]. In a randomized phase 2a trial, curcumin was combined with FOLFOX and found to be safe and tolerable (primary outcome) in 28 patients with mCRC. The study found no statistically significant differences in PFS (HR 0.57; 95% CI, 0.24–1.36; *p* = 0.2). However, there was a significant difference in OS in the intention-to-treat population (HR 0.339; 95% CI, 0.141–0.815; *p* = 0.016) [80]. 

Other potential targeted therapies against the NF-κB pathway are still in the early phases of development. Several phase 1 trials are currently underway to investigate the potential use of protein arginine methyltransferase 5 (PRMT5) inhibitors in CRC. PRMT5 is an upstream protein that regulates the NF-κB pathway [81]. PMRT5 knockout murine models are unable to produce SCs and lead to lethality [82], demonstrating the role of PMRT5 in cell stemness. PRMT5 inhibitors have shown biological activity in cancer cell lines and animal models [83]. Strategies to develop new PMRT5 inhibitors and utilize existing FDA-approved drugs to target this molecular pathway are ongoing [84,85]. 

IκB kinase alpha, a key regulator of the NF-κB pathway, has been identified as a potential target in CRC. The selective IKβ inhibitor BIX02514 has demonstrated preclinical activity in CRC cell lines by modulating the NF-κB pathway [86]. 

Other FDA-approved therapies, such as glucocorticoids, dasatinib (a tyrosine kinase inhibitor), and guratimod (an anti-rheumatic drug), have also been suggested to inhibit the NF-κB axis in cancer and may have a potential role in future development for CRC patients, revised in [72]. 

### 2.7. JAK/STAT Pathway

Janus kinase (JAK) is a family of non-receptor cytoplasmic tyrosine kinases that includes four members: JAK1, JAK2, JAK3, and TYK2. Its clinical role in myeloproliferative diseases is well-established and various JAK/STAT inhibitors are approved for treatment [87]. 

The JAK/STAT pathway contributes to CRC by supporting the survival and proliferation of cancer SCs in the inflammatory tumor microenvironment and by inducing the EMT [88]. Therapies targeting JAK/STAT signaling can both inhibit the transformation of normal SCs to CSC and promote the dedifferentiation of cancer cells to a normal stem-like state.

JAK/STAT-targeted therapies have shown promise in preclinical studies on cell lines and tumor models [88] and their impact is under active clinical investigation. However, these trials have produced conflicting results with even early termination or no published findings. 

On one hand, in a phase 3 clinical trial for pretreated mCRC patients [89], napabucasin, a STAT3 inhibitor, did not show any difference in OS compared to placebo in patients with refractory mCRC. Nevertheless, in patients with positive pSTAT3 expression, the OS was longer with napabucasin (5.1 vs. 3.0 months; HR 0.41; *p* = 0.0025). Recently, the combination of napabucasin and FOLFIRI in second-line therapy did not improve OS compared to FOLFIRI alone [90]. On the other hand, napabucasin can be safely combined with panitumumab in heavily pretreated RAS wildtype (WT) mCRC patients, showing encouraging anti-tumor activity in terms of DCR (52.1%), PFS (9.14 months), and OS (39.43 months), including 64.6% of patients who were previously treated with an anti-EGFR agent [91]. In a phase 1/2 trial the combination of napabucasin with pembrolizumab showed an ORR of 50% and 10% in mCRC with MSI-H and MSS, respectively. The DCR was 45% and the median response duration was 9 months in MSS mCRC [92]. 

Additionally, in a phase 2 trial for refractory mCRC patients, the addition of ruxolitinib (a selective JAK 1 and 2 inhibitor) to regorafenib did not result in any benefit in PFS or OS [93]. 

Finally, in a phase 1 first-in-human study with OPB-31121, a STAT3 inhibitor, Do Youn et al. observed tumor shrinkage in only two out of eighteen patients with solid tumors, both of whom had CRC [94]. 

Clinical trials are currently underway to determine the efficacy of JAK/STAT inhibitors in treating CRC and other solid tumors. Results are expected from inhibitors such as itacitinib (NCT02646748) and other STAT inhibitors (NCT03647839, NCT02983578).

### 2.8. DNA Damage Pathway

CSCs have a strong ability to repair DNA damage (DD) through multiple DNA damage response (DDR) pathways [95]. Poly ADP ribose polymerase-1 (PARP-1) mediates multiple DD repair mechanisms and it is highly expressed in CRC-SCs [96,97]. In CRC-SCs, PARP1 and other components of homologous recombination repair factors, such as RAD51 and MRE11, cooperate with the ATR-CHK1 axis to control replication stress and mitosis [98]. There are four PARP inhibitors (PARPis) that have received FDA approval for treating breast, ovary, pancreas, and prostate cancers, although in different clinical and biomarkers scenarios. PARPis are most effective in tumors associated with inability to repair DNA damage [99], including inherited or acquired mutations in BRCA1, BRCA2, or PALB2 and homologous recombination deficiency (HRD). In CRC, the mutation rates for genes involved in DDR, in HRD, and BRCA1/2 are 21%, 13–15%, and 1–8%, respectively [100]. Notably, at least one HRD gene mutation was present in over 75% of MSI-H patients and 9.5% of MSS patients [101].

In preclinical models, PARPis may enhance the cytotoxic effects of DNA-damaging chemotherapeutic agents [102]. The combination of 5-FU and veliparib (ABT-888) enhances CRC-SC death by increasing DNA damage accumulation and simultaneously inhibiting the mismatch repair (MMR) system in MMR-proficient cells [102]. Niraparib enhanced the cytotoxic activity of irinotecan in both MSI-H and MSS CRC, including in vitro and xenograft models. However, even the most sensitive CRC cell lines showed an EC50 of niraparib that was more than 10 times higher than that of BRCA-deficient cell lines [103]. Results from clinical trials of PARPis in mCRC included veliparib, olaparib, and neliparib.

Several clinical trials have evaluated the combination of veliparib with irinotecan. However, this treatment combination showed limited safety and poor tolerability in trials of escalating doses. The most common toxicities observed were diarrhea, GI toxicities, and neutropenia [104]. In a phase 2 randomized clinical trial [105], veliparib was compared to placebo, each with FOLFIRI +/− bevacizumab. There were no significant differences in any of the outcomes. The median PFS, which was the primary objective, was 12 months for veliparib and 11 months for placebo (HR, 0.94; 95% CI, 0.60–1.48). The median OS was 25 months for veliparib and 27 months for placebo (HR 1.26; 95% CI, 0.74–2.16). The response rate was 57% for veliparib and 62% for placebo. The veliparib group had significantly higher frequencies (*p* < 0.05) of anemia (39% vs. 19%, *p* = 0.019) and neutropenia (66% vs. 37%, *p* = 0.001). 

Veliparib plus temozolomide was evaluated in a single-arm phase 2 study including 75 heavily pretreated mCRC patients. The primary endpoint was met with a DCR of 25%. In addition, 4% of patients achieved a PR with a median PFS of 1.8 months and a median OS of 6.6 months. Patient inclusion was not based on DDR status. There was a suggestion of worse outcomes for patients with MSI-H tumors [106]. 

A phase 1 study was conducted to evaluate the safety and tolerability of olaparib in combination with irinotecan [107]. Olaparib was administered continuously or intermittently. Continuous olaparib and irinotecan was associated with high toxicities. Intermittent olaparib was better tolerated, although at reduced doses. Common toxicities included fatigue, anorexia, diarrhea, nausea, vomiting, neutropenia, thrombocytopenia, and abdominal pain. Nine patients had SD as the best response, two from continuous olaparib and seven from an intermittent schedule (median duration: 7.4 months; range: 4 to 13 months). The authors considered this combination to be of low interest for further clinical development. 

In a phase 2 study, olaparib in monotherapy was not effective in terms of response or PFS in patients with refractory mCRC (n = 33), regardless of MSS (n = 20) or MSI-H (n = 13) status. Only five patients had SD. Nausea, fatigue, and vomiting were the most commonly reported treatment-emergent adverse events [108]. 

The phase 3 LYNK-003 study was recently halted after enrolling 309 patients, short of its target enrollment of 525 patients [109]. The study aimed to compare olaparib alone or olaparib plus bevacizumab with 5-FU plus bevacizumab as standard maintenance therapy in patients with advanced CRC who have not progressed on first-line FOLFOX plus bevacizumab. At the prespecified interim analysis for PFS, the effectiveness of olaparib as a single therapy and in combination with bevacizumab compared to the control group did not meet the criteria for continuation as determined by the Data Monitoring Committee (DMC). Therefore, both experimental arms were discontinued (NCT04456699).

A phase 2 study assessed the safety and effectiveness of combining niraparib and panitumumab in previously treated patients with RAS wildtype (WT) mCRC [110]. The primary endpoint was the clinical benefit rate (CBR), defined as objective response and SD. Prior therapy with EGFR inhibitors (cetuximab or panitumumab) was not allowed. However, patients who had received first-line chemotherapy with oxaliplatin and had an objective response or SD for at least 4 months were eligible for the trial. The ORR was 25%, while the CBR was 83.3%. Anemia dermatitis, oral mucositis, hypertension, and neutropenia were the most common AEs (NCT03983993). 

Although preclinical studies show PARPis activity in CRC models, clinical trials have failed to demonstrate efficacy in patients with mCRC cancer. To improve the effectiveness of inhibiting DNA repair mechanisms against mCRC, several approaches can be taken: (i) using new drug combinations, (ii) developing new drugs that target different repair mechanisms, and (iii) selectively targeting patients with tumors that have altered DNA repair mechanisms.

A recent study screened a drug library in a collection of primary CRC-SCs isolated from patients [111]. The study identified prexasertib (LY2606368) as an anti-CSC agent that acts in in vitro and in vivo models. Prexasertib (LY2606368) is a small ATP-competitive selective inhibitor of CHK1 and CHK2. The most sensitive CSCs to Chk1 inhibitor showed an ongoing DNA replication stress (RS) response and were p53 deficient and hyperdiploid. This response was independent of RAS mutational status. The safety and recommended phase 2 dose of the CHK1 inhibitor prexasertib have been defined in a phase 1 trial. Hematologic toxicity was the most frequent AE and was dose limiting. The ORR was 4.9% and 12.5% in combination with cetuximab and 5-fluorouracil, respectively [112].

There is a need for biomarkers to improve the prediction of response to therapy with inhibitors of DNA repair pathways in cohorts of patients with CRC. These biomarkers would include mutations in known DNA repair genes as well as genetic alterations and genomic rearrangements indicating a repair-defective phenotype. Currently, a phase 2, open-label, multicentric, single arm trial (PEMBROLA, NCT05201612) is evaluating the response to olaparib in combination with pembrolizumab specifically in patients with mCRC with HRD. HRD has been defined as the presence of a BRCA deleterious mutation and/or a low RAD51 score (cut-off <10%) [113].

### 2.9. Hippo-YAP Pathway

The Hippo pathway consists of two main domains: the core serine/threonine kinase domain, which is modulated by a variety of upstream signals, and the transcriptional module, which is responsible for downstream target gene expression [114,115]. The core kinase domain comprises MST1/2 (Hippo or Hpo), SAV1, and LATS1/2. The transcriptional module includes YAP and TAZ. The Hippo pathway is initiated by the activation of MST1/2 or MAP4K proteins. When the kinase cascade is inactive, nuclear YAP/TAZ binds to the transcriptional enhanced associate domain (TEAD) family of transcription factors (TEAD1–4), which activate the expression of pro-proliferative and survival-enhancing genes. The Hippo pathway regulates CSCs and contributes to tumor initiation and progression [116]. Depletion of TAZ significantly decreased tumor-seeding ability. In CRC cell models, constitutive YAP activation reinforced the expression of the stemness biomarkers and regulators ALDH1A3, LGR5, and OCT4. The four main components of the Hippo pathway, MST1, LATS1/2, YAP, and TAZ, have also been shown to play a role in the development of chemoresistance, such as to 5-fluoruracil and oxalipatin [117,118]. 

There is evidence for crosstalk between the Wnt/β-catenin and Hippo-YAP signaling pathways [119]. Hippo pathway signaling is deactivated epigenetically by YAP overexpression, deletion (MST2), or gene silencing (MST, LATS, and RASSF1). In addition, KRAS and YAP1 converge on the transcription factor FOS and activate a transcriptional program involved in regulating the EMT [120]. However, in some CRC models, YAP plays a suppressive role in tumor growth. Cheung et al. [115] demonstrated that the Hippo kinases LATS1/2 and MST1/2, which inhibit YAP activity, are necessary to maintain Wnt signaling and canonical stem cell function. Loss of LATS1/2 or overexpression of YAP was sufficient to suppress tumor growth in organoids, patient-derived xenografts, and mouse models of primary and mCRC.

Currently, the development of potential drugs targeting the Hippo pathway focuses mainly on the activity and expression of the Hippo core kinase, or the expression levels of downstream YAP/TAZ, and the interactions between YAP/TAZ and TEAD. However, there are still few clinical trials with Hippo/YAP pathway modulators in cancer, including CRC. Some of the first trials include TEAD inhibitor drugs. Patients with somatic NF2 mutations and other alterations of the Hippo-YAP1 pathway are included [121]. In CRC, there are limited data available, although it is estimated that sequence variations in NF2 may be less than 5%. Expression of NF2/Merlin was reduced in CRC cells as compared with adjacent non-cancerous cells [122]. 

Another first-in-human, phase 1a/1b open-label [123] study evaluated SW-682 in patients with advanced solid tumors with or without Hippo pathway alterations. This included solid tumors with NF2 mutations and other Hippo pathway sequence variations (FAT1, LATS1/2) or YAP fusions (NCT06251310T).

## 3. Challenges in Targeting Colorectal Cancer Stem Cells

Surgery followed by adjuvant chemotherapy (ACT) for patients with stage III and high-risk stage II colon cancer is the standard of care for patients with non-metastatic disease. ACT with fluoropyrimidines and combination with oxaliplatin increase cure rates. In a fraction of patients, they are able to eradicate occult disseminated CSCs [124]. The treatment for unresectable metastatic CRC is based on systemic therapy (cytotoxic chemotherapy, biologic therapy such as antibodies to VEGF and EGFR growth factors and their combinations, and tyrosine-kinase inhibitor). Immunotherapy is used in the subset of MSI-H tumors. Currently used drugs target rapidly dividing cells and, while often effective at debulking tumor mass, these agents have largely failed to cure disease in the metastatic setting. It has been suggested that CSCs possess characteristics of tumor dormancy and inherent resistance to cytotoxic drugs, which may result in disease relapse after initial therapy involving surgery and adjuvant chemotherapy and in metastases and chemoresistance. Tailored approaches against CSCs could potentially be useful at any of these moments, from the presence of minimally disseminated but quiescent tumor disease, metastases chemosensitive to initial treatments, and even the situation of clinical progression with refractory disease. Currently, there are several therapeutic strategies under preclinical and clinical evaluation that could specifically target and eliminate CSCs. Table 1 shows different randomized trials that evaluate the efficacy of different anti-target drugs with potential activity against SCs in CRC. It is important to note how the various clinical trials targeting stemness have been developed in different evolutionary stages and clinical scenarios of mCRC. Determining the appropriate drugs and timing of administration is a crucial factor for the potential effectiveness of this therapy against CRC-SCs. 

Notably, CSCs represent a small percentage (0.3% to 2.2%) of all cancer cells in CRC [125]. There is evidence that systemic spread can occur early in CRC [126]. Cells with functional and phenotypic properties of SCs have been isolated from primary colorectal tumors, metastases, and blood [127]. Although the expression of distinct stem cell markers in both tumor tissue and peripheral blood has been correlated with prognosis, we have no standardized and validated method in the clinic to enumerate or characterize CRC-SCs in real time [7]. The majority of studies have included patients with mCRC without selection based on molecular markers that could indicate the activity of the different signaling pathways. The identification of SC signatures specific to each patient could guide personalized treatment strategies. 

CRC-SCs display a heterogeneous and dynamic phenotype that is influenced by mutations in epithelial cells and by extrinsic signals from the microenvironment [128]. Moreover, this plasticity allows SCs to adapt to the selective pressure that can be induced, for example, by anti-neoplastic treatment. The lack of efficacy of the different anti-target therapies in eradicating CRC-SC may be partly explained by these characteristics. 

To address these challenges, the use of single-cell omics technologies offers a promising avenue to unravel the complex heterogeneity and plasticity of CSCs. By providing high-resolution insights into the molecular signatures of individual cancer cells, single-cell omics can significantly enhance our ability to identify specific biomarkers and therapeutic targets unique to CSCs. Moreover, single-cell analysis may facilitate the real-time monitoring of therapeutic responses and CSC dynamics, offering new approaches for managing cancer progression and resistance mechanisms [129]. 

In the clinical context, addressing the challenge of eradicating CSCs and preventing tumor recurrence while preserving normal SCs, which are crucial for tissue regeneration and homeostasis, requires a multifaceted approach. One such challenge is the inherent plasticity of CSCs, which allows them to alternate between stem-like and differentiated states. Furthermore, the identification of reliable biomarkers for CSCs remains a significant barrier to effective clinical management. Taking advantage of these cutting-edge technologies, researchers and clinicians can move closer to designing effective therapeutic strategies that address the unique challenges posed by CSCs in CRC. While the eradication of SCs has been hypothesized as a possible cure for solid tumors, including CRC [130], there is still considerable progress yet to be made.

## Figures and Tables

**Figure 1 ijms-25-06220-f001:**
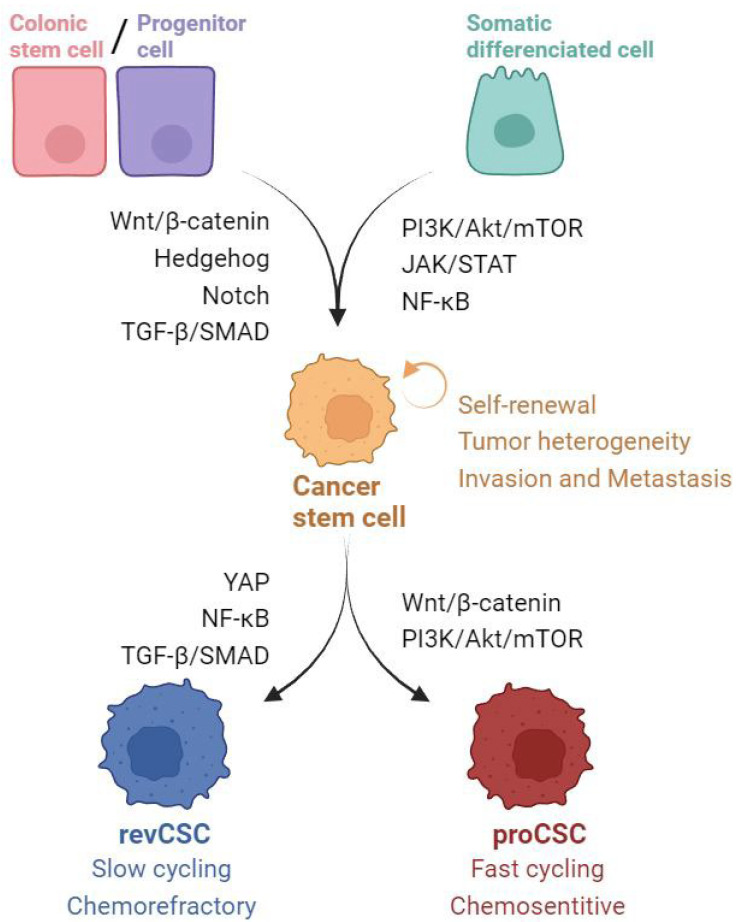
Signaling Pathways and Their Roles in Colorectal Cancer Stem cell (CRC-SC) hallmarks. These pathways collectively contribute to the intricate regulatory network that supports the malignancy and plasticity of CSCs and highlights potential therapeutic targets.

**Table 1 ijms-25-06220-t001:** Phase 2 and 3 randomized clinical trials evaluating targeted molecular pathways associated with colorectal cancer SCs.

Trial	Pathway	Phase	CRC Patients	Intervention	Comparator	Significant Results	Reference
CORRECT	Wnt/β-Catenin	3	Pretreated	Regorafenib	Placebo	Regorafenib mOS, 6.4 months. Placebo 5 m. HR 0.77; one-sided *p* = 0.0052	[18]
COAST	Wnt/β-Catenin	3	Following curative resection of liver metastases after completion of planned chemotherapy	Regorafenib	Placebo	Only 25 patients randomized. SAEs were observed in 14.29% with regorafenib	[24]
NCT01830621	JAK/STAT	3	Pretreated	Napabucasin	Placebo	OS did not significantly differ. In pSTAT3-positive patients, napabucasin showed longer survival: median 5.1 months (95% CI 4.0–7.5; HR 0.41, 0.23–0.73, *p* = 0.0025)	[89]
LYNK-003	DNA Damage Response	3	Advanced CRC patients who had not progressed on first-line FOLFOX plus bevacizumab	Olaparib/ Olaparib plus bevacizumab	5-FU + bevacizumab	The study was terminated for futility after enrolling 309 patients	[109]
NCT00985192	PI3K/Akt/mTOR	2	Pretreated	Sonolisib (PX-866) and cetuximab	Cetuximab	No PFS superiority. The treatment-related toxicity was higher in the combination arm	[61]
NCT02278133	PI3K/Akt/mTOR	2	BRAF V600E-mutated mCRC	Alpelisib, encorafenib, and cetuximab	Encorafenib and cetuximab	The combination of cetuximab, encorafenib, and alpelisib resulted in a median PFS 5.4 months (HR 0.69; *p* = 0.064). ORR 27% in 52 patients	[67]
NCT01490996	NF-κB	2a	Pretreated	Curcumin + FOLFOX	FOLFOX	No statistically significant differences in PFS (HR 0.57; *p* = 0.2). However, there was a significant difference in OS in the ITT population (HR 0.339; *p* = 0.016)	[79]
NCT02119676	JAK/STAT	2	Pretreated	Ruxolitinib + regorafenib	Regorafenib	The combination did not improve OS or PFS	[93]
NCT02305758	DNA Damage Response	2	Pretreated	Veliparib + FOLFIRI	Placebo + FOLFIRI	No significant differences in PFS or OS. The veliparib group had significantly higher frequencies of anemia and neutropenia	[105]
NCT00636610	Hedgehog	2	Pretreated	Vismodegib + FOLFOX/FOLFIRI + bevacizumab	Placebo + FOLFOX/ FOLFIRI + bevacizumab	The overall response rates for placebo-treated and vismodegib-treated patients were 51% (90% CI: 43–60) and 46% (90% CI: 37–55), respectively	[32]

## Data Availability

No new data were created.

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
