# Peer review of "Targeted Treatment against Cancer Stem Cells in Colorectal Cancer"

_ijms, 2024, doi:10.3390/ijms25116220_

Round 1
Reviewer 1 Report
Comments and Suggestions for Authors
A very extensive and detailed paper that proposes and synthesizes the main studies of the field aimed at overcoming the resistance of colon cancer cells and preventing their development. First of all, it is necessary to inform readers How are CSCs obtained and studied? It should be specified whether these cells are taken from peripheral blood or tumour tissue or lymph nodes. Since most of the readers are not expert biologists but many are clinicians, it is necessary to give some details on how CSCs are acquired and how they are selected and studied. In the introductory part I would add a mention of the discovery of these CSCs and how they were highlighted and noticed. In the tables above, it would be useful to correlate the results with the percentages of Csc present or believed to be present in patients.
In other words, did the reported negative studies fail because there were no evaluable CSCs, because they were not recognized, because of unrecognized growth factors? Another question that readers ask themselves is: are CSCs all the same or does each one express a different genome from the other? I would reduce the number of bibliographic citations to the last 3 years and reduce the longer chapters by syneticizing them. As it is, the work is too long and tedious
Comments on the Quality of English LanguageEnglish well written and correct
Author Response
We appreciate the comments and suggestions
We have restructured the paper and the number of references. A more detailed introduction about the origin, source and methods of stem cell studies is included. The limitations of counting SCs in clinical trials are revealed. Finally, a comprehensive discussion about the challenges for SCs target are provided.
Reviewer 2 Report
Comments and Suggestions for Authors
This manuscript is a review of clinical trials that focused on an important signaling pathway in colorectal cancer (CRC) and evaluated the efficacy of its molecularly targeted therapies. Such a review is needed and should be updated in the context of the increasing popularity of molecularly targeted therapies. This is well-written and seems to summarize well the current state of them in this area. Just a couple of questions and suggestions are listed below.
Specific points)
1. Page 2, lines 52-55; the meaning of this seems unclear. It seems to imply that CSCs are supplied by normal stem cells?
2. Figure 1; The information is somewhat lacking. Personally, I would like to see a schematic diagram of the interrelationships and mechanisms of action of each target molecule.
3. The meaning of some sentences is unclear, probably due to typing errors. The following are examples of those.
Page 7, line 256; TGF-B antibodies?
Page 8, line 279; PD-L1.
Page 9, line 336; T A seminal work?
Comments on the Quality of English LanguageAs for the quality of the English language, a few expressions were found to be somewhat unclear.
Author Response
We appreciate the Comments and Suggestions
We incorporate the Specific points indicated.
1. Page 2, lines 52-55; the meaning of this seems unclear. It seems to imply that CSCs are supplied by normal stem cells?
We have rewritten and clarified the phrase. Lines 56-58: “ CRC-SCs have two main sources: SCs that have undergone malignant transformation through genetic or epigenetic alterations, and differentiated epithelial non-SCs that acquire stemness through dedifferentiation [6, 8-11]”.
2. Figure 1; The information is somewhat lacking. Personally, I would like to see a schematic diagram of the interrelationships and mechanisms of action of each target molecule.
We have designed a new, clearer figure. Figure 1.
3. The meaning of some sentences is unclear, probably due to typing errors.
We have corrected the errors.